# Comparison of Resistance Spectra after First and Second Line Osimertinib Treatment Detected by Liquid Biopsy

**DOI:** 10.3390/cancers13122861

**Published:** 2021-06-08

**Authors:** Balázs Jóri, Stefanie Schatz, Len Kaller, Bettina Kah, Julia Roeper, Hayat O. Ramdani, Linda Diehl, Petra Hoffknecht, Christian Grohé, Frank Griesinger, Markus Tiemann, Lukas C. Heukamp, Markus Falk

**Affiliations:** 1Institut für Hämatopathologie Hamburg, Fangdieckstraße 75A, 22547 Hamburg, Germany; jori@hp-hamburg.de (B.J.); schatz@hp-hamburg.de (S.S.); kah@hp-hamburg.de (B.K.); mtiemann@hp-hamburg.de (M.T.); heukamp@hp-hamburg.de (L.C.H.); 2Lung Cancer Network NOWEL, 26129 Oldenburg, Germany; len.kaller@semmelweis-hamburg.de (L.K.); julia.roeper@uni-oldenburg.de (J.R.); hayat.oum.el.kheir.ramdani@uni-oldenburg.de (H.O.R.); Petra.Hoffknecht@niels-stensen-kliniken.de (P.H.); Christian.Grohe@jsd.de (C.G.); Frank.Griesinger@Pius-Hospital.de (F.G.); 3Asklepios Campus Hamburg, Semmelweis University, Lohmühlenstraße 5, 20099 Hamburg, Germany; 4Department of Hematology and Oncology, Pius-Hospital Oldenburg, Georgstraße 12, 26121 Oldenburg, Germany; 5Department of Internal Medicine-Oncology, University of Oldenburg, Georgstraße 12, 26121 Oldenburg, Germany; 6Institute of Experimental Immunology and Hepatology, University Medical Center Hamburg Eppendorf, Martinistraße 52, 20246 Hamburg, Germany; li.diehl@uke.de; 7Department of Thorax Oncology, Niels-Stensen-Kliniken, Franziskus-Hospital Harderberg Alte Rothenfelder Straße 23, 49124 Georgsmarienhütte, Germany; 8Department of Pneumology, Evangelische Lungenklinik Berlin, Lindenberger Weg 27, 13125 Berlin, Germany

**Keywords:** liquid biopsy, lung cancer, routine diagnostics, osimertinib resistance

## Abstract

**Simple Summary:**

Since the recent approval of osimertinib, a third generation tyrosine kinase inhibitor (TKI) targeting *EGFR* in non-small cell lung cancer (NSCLC), tracing the resistance mechanisms that yield to failure of osimertinib has become of interest. As the spectrum of osimertinib-resistance related genomic alterations appears significantly more diverse compared to first and second generation TKI, comprehensive, and preferably non-invasive molecular diagnostic methods are required for the detection of resistance mechanisms. In this study, we present molecular results of 56 NSCLC patients during disease progression on first and second line osimertinib treatment using a hybrid capture (HC) next generation sequencing (NGS) based liquid biopsy approach. We show examples of polyclonal resistance development which leads to the presence of multiple resistance mechanisms in the same patient, and highlight the clinical utility of HC NGS over single gene testing.

**Abstract:**

Since 2009, several first, second, and third generation *EGFR* tyrosine kinase inhibitors (TKI) have been approved for targeted treatment of *EGFR* mutated metastatic non-small lung cancer (NSCLC). A vast majority of patients is improving quickly on treatment; however, resistance is inevitable and typically occurs after one year for TKI of the first and second generation. Osimertinib, a third generation TKI, has recently been approved for first line treatment in the palliative setting and is expected to become approved for the adjuvant setting as well. Progression-free survival (PFS) under osimertinib is superior to its predecessors but its spectrum of resistance alterations appears significantly more diverse compared to first and second generation *EGFR* TKI. As resistance mechanisms to osimertinib are therapeutically targetable in some cases, it is important to comprehensively test for molecular alterations in the relapse scenario. Liquid biopsy may be advantageous over tissue analysis as it has the potential to represent tumor heterogeneity and clonal diversification. We have previously shown high concordance of hybrid capture (HC) based next generation sequencing (NGS) in liquid biopsy versus solid tumor biopsies. In this study, we now present real-word data from 56 patients with metastatic NSCLC that were tested by liquid biopsy at the time of disease progression on mostly second line treated osimertinib treatment. We present examples of single and multiple TKI resistance mechanisms, including mutations in multiple pathways, copy number changes and rare fusions of *RET*, *ALK*, *FGFR3* and *BRAF*. In addition, we present the added value of HC based NGS to reveal polyclonal resistance development at the DNA level encoding multiple *EGFR* C797S and *PIK3CA* mutations.

## 1. Introduction

*EGFR* mutations predominantly occur in non-squamous NSCLC with a frequency of about 10–15% in the western world. These patients benefit from *EGFR* TKI, however, remission is typically followed by relapse after a median of 9–18 months [1,2]. Resistance in response to first- and second-generation *EGFR* TKI (i.e., gefitinib, erlotinib, and afatinib) is most commonly conferred by the gatekeeper mutation T790M [3]. Other alterations like *MET* gene amplification have been reported in up to 15% of patients and provided a rationale for the addition of *MET* inhibitors such as crizotinib or tepotinib to overcome resistance [4,5].

The third-generation TKI osimertinib was originally approved for second line treatment since it showed activity for both the classical *EGFR* mutations and T790M, leading to the concept of sequential TKI therapy, including a switch to osimertinib upon progress and T790M positivity. After the recent approval of osimertinib for first line treatment, the concept of sequential TKI therapy has been challenged and is still a matter of debate [6]. The “best first” strategy of oncology favors osimertinib as first line option for *EGFR* mutated NSCLC based on the overall survival (OS) benefit compared to first-generation *EGFR* TKI and the prevention of the most frequent resistance mechanism, T790M. Resistance mechanisms to osimertinib are therefore important to capture by comprehensive testing in order to identify those patients with targetable resistance mechanisms.

The most common resistance mutation after second line osimertinib affects the C797X residue in *EGFR*, the binding site for this substance [7,8,9,10]. In case C797X is occurring in cis configuration with T790M as (T790M; C797S), it renders the tumor insensitive to any currently approved *EGFR* TKI. The setting of *EGFR* C797X in trans to T790M, as (T790M); (C797S), is rare compared to cis. However, data indicate sensitivity of in trans configuration of (T790M); (C797S) to a combination first- or second-generation *EGFR* TKI together with a third-generation *EGFR* TKI [11,12]. The overall spectrum of resistance alterations in response to osimertinib is remarkably more complex than to its class members of the prior generations [6]. Fusions of *NTRK* and *ALK* have been reported as well as amplification of *ERBB2*, *EGFR* and *MET*, or point mutations leading to activation of *RAS/MAPK* or *PI3K/AKT/mTOR* pathways [10,13,14,15].

The accurate identification of such genetic alterations is crucial not just for patient management but also for advancing our understanding of treatment-induced tumor evolution. As the feasibility of repeated tissue biopsies during tumor progression is clinically problematic, non-invasive methods, such as liquid biopsy, emerged as complement to routine tissue-based diagnostics and became recommended in the new College of American Pathologists, International Association for the Study of Lung Cancer, and Association for Molecular Pathology guidelines for molecular testing of patients with metastatic NSCLC [16,17].

Hotspot testing methods, such as droplet digital PCR or enhanced-ice-COLD-PCR might allow high sensitivity in blood-based resistance testing of metastatic cancers [18,19]. However, these assays are limited to covering only narrow genomic territories and are unsuitable to reveal complex and heterogeneous resistance mechanisms that are commonly occurring during osimertinib treatment [6]. In this regard, HC NGS that provides comprehensive analysis of point mutations, small InDels, copy number alterations, and gene fusions, including exonic and intronic regions of onco- and suppressor genes, from liquid biopsy samples might deliver the demanded clinical utility [20].

In this work, we highlight patients that were sequentially tested via liquid biopsy and focus on resistance to osimertinib mainly in the second line setting. We provide examples for the clinical applicability and added value of HC NGS based liquid biopsy in resistance testing in terms of known and novel genomic alterations, multiple resistance mechanisms and evidence for development of polyclonal resistance.

## 2. Materials and Methods

### 2.1. Patient Cohort

Among a cohort of over 700 patients tested via liquid biopsy in the central laboratory of North-East-West Lung Network (NOWEL), a German lung cancer network, we identified a total of 56 patients who developed progressive disease on osimertinib treatment. As part of NOWEL, plasma ctDNA was analyzed as an alternative to tissue biopsy because (1) patients refused a solid tumor biopsy, (2) tissue biopsy was not possible, or (3) tissue derived DNA was not sufficient for coverage of all targetable driver alterations. The *EGFR* mutation status at primary diagnosis had either been assessed in our institution (Institut für Hämatopathologie Hamburg, Germany) at an earlier time point or was documented as part of the NOWEL clinical documentation. Collating the resistance pattern to osimertinib, we excluded plasma samples that met the assay’s quality criteria, but tested wildtype for the initial *EGFR* driver mutation, presumably indicating limited tumor DNA shedding.

### 2.2. Liquid Biopsy

Whole blood (18 mL) was collected in two Streck Cell-Free DNA BCT^®^ (Streck, Omaha, NE, USA) and cfDNA was extracted using Qiagen’s QIAamp Circulating Nucleic Acid kit (Qiagen, Hilden, Germany). Liquid Biopsies were taken at the time of progression under osimertinib. CfDNA was subjected to HC based NGS to detect point mutations, small InDels, copy number alterations and genomic translocations (NEOliquid v1, NEO New Oncology GmbH, Köln, Germany). Details of the panel setup are included in Appendix B. In brief, after DNA extraction, adapters were ligated, and DNA was processed as described previously [21]. After enrichment, targeted fragments were clonally amplified and sequenced in parallel at an ultrahigh sequencing depth averaging 37,150× (unfiltered reads). Computational analysis was performed with NEO New Oncology’s proprietary computational biology analysis pipeline to remove sequencing artefacts and detect relevant genomic alterations in a quantitative manner (minor allele frequency ≥0.1%). Of note, due to assay limitation, in copy number variation (CNV) calling enabled only the detection of amplifications in *EGFR*, *ERBB2,* and *MET*. The detection of CNV deletions, such as in *RB1* or *TP53* was not possible.

## 3. Results

### 3.1. Patient Characteristics

The mean age of the 56 patients that received osimertinib as either first or second line *EGFR* TKI was 60 years with a female/male ratio of 64% versus 36%. Smoking status could not be assessed in 6/56 (11%) of patients, while 31/56 (55%) were reported to be never smokers, 2/56 (4%) current smokers, and 17/56 patients (30 %) former smokers. Pack years were documented for 14/17 former smokers with a median of 17py. Histomorphological classification revealed a majority of adenocarcinomas in 54/56 of the cases (96%), besides one case (2%) adeno-squamous and large-cell carcinoma, respectively. Clinical stage of disease was UICC stage IV exclusively (98%) while in 2% documentation was incomplete (Table 1). In nine of 56 patients osimertinib was administered first line, and in 47 of 56 patients as second line TKI. Of the latter, 37 of 47 patients received one first or second generation TKI prior to osimertinib and eight of 47 patients received two different TKI prior to osimertinib. In two patients therapy included rociletinib, a third generation *EGFR* TKI that did not gain approval. Two of 47 patients received a sequence of gefitinib, erlotinib and afatinib in varying order during the course of the disease before osimertinib was administered.

The most predominant *EGFR* activating mutations were the common exon 19 deletions (40/56, 71%), followed by L858R (13/56, 23%). In 3/56 (5%) patients, uncommon mutations were identified as the original driver, in two of which, concomitant *EGFR* mutations were present ((E709G; G719S), (E709V; G719C), L861Q). de novo T790M mutations in TKI naïve patients were not detected at primary diagnosis. The detected resistance spectrum to osimertinib is shown in Figure 1. A comprehensive patient specific list is provided in the supplements (Appendix A).

### 3.2. Resistance Spectrum after First Line Osimertinib Treatment

Nine patients (9/56, 16%) in our cohort received osimertinib first line. Seven plasma samples confirmed the initial driver mutation. Five of them carried common *EGFR* mutations (exon 19 deletion/L858R) as the original driver, while two showed uncommon mutations L861Q and (E709G, G719S). These driver mutations are currently being discussed as being less sensitive to *EGFR* TKI [22,23,24].

In three of seven patients treated with osimertinib first line resistance mechanisms were detected. One patient (1/7, 14%) showed a C797S resistance mutation, in another, alterations in *TP53* and *RB1* indicated potential tumor transformation to SCLC [11,25,26] although no solid tumor biopsy was available for histological confirmation. In a third patient, multiple resistance mechanisms were observed in a single plasma sample, including *MET* amplification, *EML4-ALK* fusion (v3) and *EGFR* T854A mutation. The T854A mutation was found in combination with the driver L858R mutation with similar allelic frequencies, suggesting a concomitant (L858R; T854A) mutation. In vitro, T854A in combination with L858R was described as refractory to first/second generation TKI but sensitive to third generation *EGFR* TKI [25]. In four of seven (57%) patients no on or off target resistance alteration was detectable, although one of these cases displayed a *BRCA2* mutation. Six patients (6/7, 86%) carried a *TP53* mutation. Of note, no patient receiving osimertinib first line developed a T790M mutation, nor was it present prior to treatment.

### 3.3. Resistance Spectrum after Second Line Osimertinib Treatment

A number of 47 patients received osimertinib second line, of those, six plasma samples did not reveal the initial driver alteration, indicative of limited ctDNA shedding and therefore were excluded from further analysis. Of the remaining 41 patients, 37 (90%) had developed a T790M in response to a previously administered first or second generation TKI. Four patients were switched from first/second generation TKI to osimertinib despite the absence of T790M. In 20% (8/41) of plasma samples no obvious resistance alteration to osimertinib was detectable, although the primary driver mutation was readily visible and the liquid biopsy was sent to us at the time of progression to osimertinib. Nevertheless, in these cases clinically relevant passenger mutations were detectable at the time of progression, including inactivating mutations in *ATM* (2/41, 5%) and *TP53* (6/41, 15%). Altogether, 33/41 (80%) of samples harbored at least one alteration defined as an acquired driver alteration, and therefore presumably causing resistance to osimertinib. In descending order of frequency, we detected *EGFR* C797S in 16/41 (39%) cases, followed by 12% (5/41) samples with non-C797S *EGFR* mutations including V843I, L718Q, C724S, and L792H, and one patient with (L718V, L718Q, L792H, G796S). The V843I mutation has been described as a lung cancer predisposing variant so we cannot exclude its germline origin (Variant Allelic Frequency, VAF = 82.3%) [26,27]. In 7% (3/41) of cases ctDNA showed concomitant *RB1* and *TP53* inactivating mutations, indirectly indicative of transformation to SCLC. Due to a lack of FFPE material from these patients, this however could not be confirmed histomorphologically. *EGFR* amplification occurred in 10% (4/41) of cases and *MET* amplification in 3/41 (7 %) cases, respectively. *CTNNB1* point mutations occurred in three (7%), *KRAS* mutations as well as *PIK3CA* activating mutations in two (5%), and one of each *ERBB2*, *PTEN*, *mTOR,* and *RET* mutation in one (2%) of the 41 patient samples. Furthermore, we identified gene fusions of *AGK*-*BRAF*, *RET*-*RUFY1, TACC-FGFR3,* and *DLG1*-*BRAF* (each 1/41, 2%). Reports in the literature have defined “loss of T790M” as another mechanism of resistance to third generation *EGFR* TKI [28]. In our cohort, 14/41 (34%) plasma samples were devoid of T790M after osimertinib treatment, although it was detectable at an earlier time point.

Of note, the above listed resistance alterations did not necessarily occur mutually exclusive but in combinations of up to four in a single liquid biopsy sample. 

It is still a matter of debate whether resistance mutations do preexist prior to TKI therapy in form of low frequency subclones or are secondarily acquired during the course of TKI therapy. For two patients with C797S positive plasma, FFPE tissue from primary diagnosis was available. We reevaluated the tissue derived DNA for a potential a priori presence of C797S by applying HC NGS with an limit of detection of 0.1% similar to that used for ctDNA analysis. While the initial exon 19 deletions were confirmed in these patients, the C797S mutation was not detected in the TKI naïve setting (data not shown). However, the presence of preexisting subclones below detection levels cannot be completely ruled out.

### 3.4. Presence of Diverse EGFR and PIK3CA Molecular Subclones of in Patients’ Plasma ctDNA after Second Line Osimertinib

Altogether, we detected the C797S variant in 16(39%) of 41 patients that received osimertinib as second line TKI, all of which displayed synchronous T790M in the liquid biopsy. In four of the 16 patients with detectable plasma C797S ctDNA, we detected two distinct nucleotide variations, specifically c.2389T > A and c.2390 G > C, both separately encoding for C797S. Consequently ctDNA of one patient displayed three distinct alleles (1) T790M (2) T790M + C797S (c.2389T > A) (3) T790M + (C797S c.2390 G > C) (Figure 2A). Another patient even displayed five different alleles originating from at least three different tumor subclones (1) T790M (2) C797S (c.2389T > A) (3) C797S (c.2390 G > C) (4) T790M + C797S (c.2389T > A) (5) T790M + C797S (c.2390 G > C) (Figure 2B).

We evaluated other single-nucleotide variants in that regard, assuming that our finding may represent subclonal divergence of resistance patterns beyond the protein level. Indeed we detected two activating *PIK3CA* mutations in one plasma sample, E545K (c.1633G > A. VAF = 3.81%) and E542K (c.1624G > A, VAF = 0.20%) giving rise to four subclones altogether in this specific patient (1) T790M (2) T790M + *PIK3CA* (L792H) (3) T790M + *PIK3CA* (G796S) (4) T790M + C797S (c.2389T > A) (Figure 2C).

Resistance in response to third generation TKI appears more heterogeneous than to earlier generation TKI as reflected by subclonal ramification at protein and DNA levels [6]. For example, one patient’s ctDNA in our cohort was defined by multiple resistance alterations at the time of progression on second line osimertinib, including C797S, *TACC*-*FGFR3* fusion along with the T790M from previous TKI treatment.

## 4. Discussion

With the intent to decipher resistance mechanisms in response to osimertinib, we retrospectively collected HC NGS data of liquid biopsies from 56 patients that progressed under osimertinib treatment. 

Of the seven patients treated with osimertinib first line and cfDNA testing proved successful, one patient (14%) developed a C797S resistance mutation. C797X is a common resistance mechanism against osimertinib that has been previously described especially after second line osimertinib. In our cohort, 39% of patients treated with osimertinib in second line developed C797S mutations, which is significantly higher than the 15% C797X mutation rate observed in the AURA3 trial [29]. In AURA3, however, patients were tested for resistance alterations that either progressed or treatment was discontinued for other reasons. Thus, the patients in our cohort may be more clearly defined by clinical progression at the time of liquid biopsy taking. This possibility is strengthened by the observation that in AURA3, 60% of patients had no known resistance alteration versus merely 20% in our cohort [29]. It could further be speculated that different sensitivity levels of liquid biopsy might be one of the reasons for the difference. The assay we used was validated with a lower limit of detection of 0.1% for single nucleotide variants, while the performance specifications of the assay used in AURA3 indicates a range of 0.05–0.25% for single nucleotide variants [29].

In one patient, concomitant *TP53* and *RB1* mutations suggested SCLC transformation. Despite the limitation that we could not confirm this finding histomorphologically, it will be interesting to see whether combination of osimertinib with chemotherapy (FLAURA2 trial) will prevent such resistance. Furthermore, studies are underway, combining *EGFR* TKI with VEGF inhibitors and it will be instructive to observe its potential impact on resistance pattern [30].

Regarding the second line osimertinib subgroup, all except one C797S mutated patients showed additional resistance mechanisms. Multiple resistance mechanisms highlight the importance of complex and well-designed molecular diagnostic assays and might be one explanation for primary resistance to osimertinib in second line.

Among the additional alterations, we detected gene translocations including *EML4-ALK* (v3), *TACC3-FGFR3*, *RET-RUFY1,* and two different *BRAF* fusions. As detection of fusions by RNA-based assays from liquid biopsy is challenging, DNA-based HC NGS provides a valid alternative. Further, we detected activating mutations in *KRAS*, *ERBB2*, *RET,* and *CTNNB1*, presumably conferring a new oncogenic driver in the respective tumor. The *PIK3CA/PTEN/mTOR* axis was mutated in four patients, including the hotspot mutation E545K. In one patient, two simultaneous *PIK3CA* mutations were detectable, suggesting polyclonal resistance development. CtDNA of four patients displayed concomitant *RB1* and *TP53* mutations indicative of SCLC transformation, however no solid tumor biopsy was available for histological confirmation. Amplifications of *MET* were detected in 7% of patients treated with second line. In the FLAURA trial 15% of patients treated with first line osimertinib developed *MET* amplification, comparing well to our data, although patient number in our cohort was limited [31]. In our second line osimertinib group, 7% of patients displayed *MET* amplification, roughly half the frequency of AURA3 with 19% *MET* amplification [29]. 

Irrespective of the line of treatment, testing only for the C797X hotspot would have detected merely half of the actual resistance alterations compared to HC NGS. Considering the potential off-label use of targeted therapeutic options and available clinical trials, we identified another 19 % of patients with druggable alterations such as *BRAF-* and *FGFR3*-fusions and clinically relevant mutations in *RET*, *ERBB2*, *PIK3CA*, and *MTOR*. These findings underline the necessity of biomarker-directed studies such as the ORCHARD trial, allowing for individual molecular stratification based on the type resistance alteration post osimertinib progression [32].

“Loss of T790M“ is an often cited concept of osimertinib resistance and a recent meta-analysis indicates prolonged PFS in cases where T790M persists [28]. In our cohort, T790M did not occur in patients treated with osimertinib first line. In contrast, about 90% of patients in our cohort did develop T790M upon treatment with *EGFR* TKI of the first or second generation. After these patients were switched to osimertinib, 34% had “lost” T790M, since it ceased to appear in the liquid biopsy. The T790M point mutation increases ATP affinity of the *EGFR* ATP binding pocket and confers the most common type of resistance in response to ATP equivalent TKI of the first and second generation [33]. In contrast, osimertinib overcomes T790M through increased hydrophobic interactions with Met790 versus Thr790 and an improved rate of covalent bond formation via more effective positioning of the acrylamide residue [34]. Suppression of T790M positive tumor cell clones along with synchronous appearance of resistance alterations in response to osimertinib therefore appears to be a compelling mechanism of tumor evasion to selective TKI pressure. The term “loss of T790M” is however suggestive of genetic clearance of T790M on a previously mutated allele. In that sense the term may be misleading because mechanistically, suppression of the T790M containing tumor subclone may be the underlying reason. Our ctDNA analyses at the time of osimertinib progression revealed undetectable levels of T790M in a significant portion of cases, presumably reflecting lack of DNA shedding of T790M positive subclones.

Our data supports an emerging picture that osimertinib may promote molecular diversification to a level beyond that of TKI of the first and second generation. 

Intriguingly, in four patients we detected development of both *EGFR* C797S versions (c. [2389T > A]; [2390G > C]), indicative of a subclonal ramification even beyond the protein level that may mirror the effect of a more selective TKI pressure. We want to point out, however, that one of these patients received rociletinib, another third generation TKI prior to osimertinib, so at least in this specific case the resistance alteration cannot clearly be attributed to either TKI. Plasma ctDNA analyses are by nature snapshots of a dynamic and heterogeneous tumor development. In the absence of longitudinal tissue biopsies including metastases, it is not possible to trace back the respective time points where clonal branching had occurred.

A fundamental debate is still ongoing whether resistance emerges by selection of preexisting low level subclones or by de novo acquisition of mutated alleles [35]. We therefore retrospectively sequenced solid tumor biopsies of two patients whose ctDNA revealed presence of nucleotide changes that lead to C797S. The comparison of FFPE with liquid samples somewhat favored the concept of de novo acquisition by not containing C797S mutations or the *FGFR3* fusion, however, the coverages in the FFPE samples did not allow us to draw distinct conclusions in this regard (data not shown).

First and second generation *EGFR* TKI act as non-covalently binding competitive ATP equivalents and, therefore, target *EGFR* irrespectively of wildtype or mutated configuration. T790M sterically hinders binding of these TKI to the ATP-binding site of *EGFR* [36]. Osimertinib covalently binds to *EGFR* residue C797X and therefore specifically targets activating *EGFR* mutations including T790M, but to a lesser extent wildtype *EGFR*. In vitro, osimertinib acts more effective, as represented by lower IC50 values than most first and second generation TKI with minimal off-target effects translating into less adverse events in patients [37]. This has led to the concept of sequential therapy, starting with a TKI of first or second generation and upon relapse and confirmation of T790M adjusting second line treatment to osimertinib to achieve prolonged overall TKI effectivity. Based on FLAURA trial, osimertinib has been approved for first line treatment with a better outcome (PFS and OS) compared to erlotinib or gefitinib [2]. Especially the improved OS in the FLAURA trial has somewhat settled the discussion of sequence therapy versus “best first”. As it is still unclear which patients might develop T790M and which therefore might benefit from a sequence strategy, the standard of care has moved to osimertinib in first line. However, concepts of early detection of T790M and early treatment with osimertinib are currently investigated in clinical trials and it will be important to see, whether these concepts will lead to an improved OS for the overall population in comparison to osimertinib first line.

As osimertinib is about to receive expanded approval for the adjuvant setting, increasing numbers of NSCLC patients of all stages with an inherently better prognosis and longer therapeutic interval will be treated with osimertinib in the future. Other TKI are currently not approved in the adjuvant setting, hence, the use of osimertinib will be without alternative for early stage tumors when it comes to TKI treatment. Studies are urgently needed to assess the potential risk of osimertinib to promote fast subclonal tumor evolution inducing diversification of resistance.

## 5. Conclusions

Resistance to osimertinib imposes a significant challenge as post-osimertinib options are limited as of now. Our study confirms the complexity and wide spectrum of resistance mechanisms, such as rare gene fusions, copy number alterations, cellular transformation, and activation of alternate pathways after osimertinib treatment. The applied HC NGS liquid biopsy proved clinical utility by identifying actionable alterations. These data highlight the importance of longitudinal and comprehensive molecular testing and might be crucial to decipher third generation TKI resistance mechanisms to guide subsequent therapies.

## Figures and Tables

**Figure 1 cancers-13-02861-f001:**
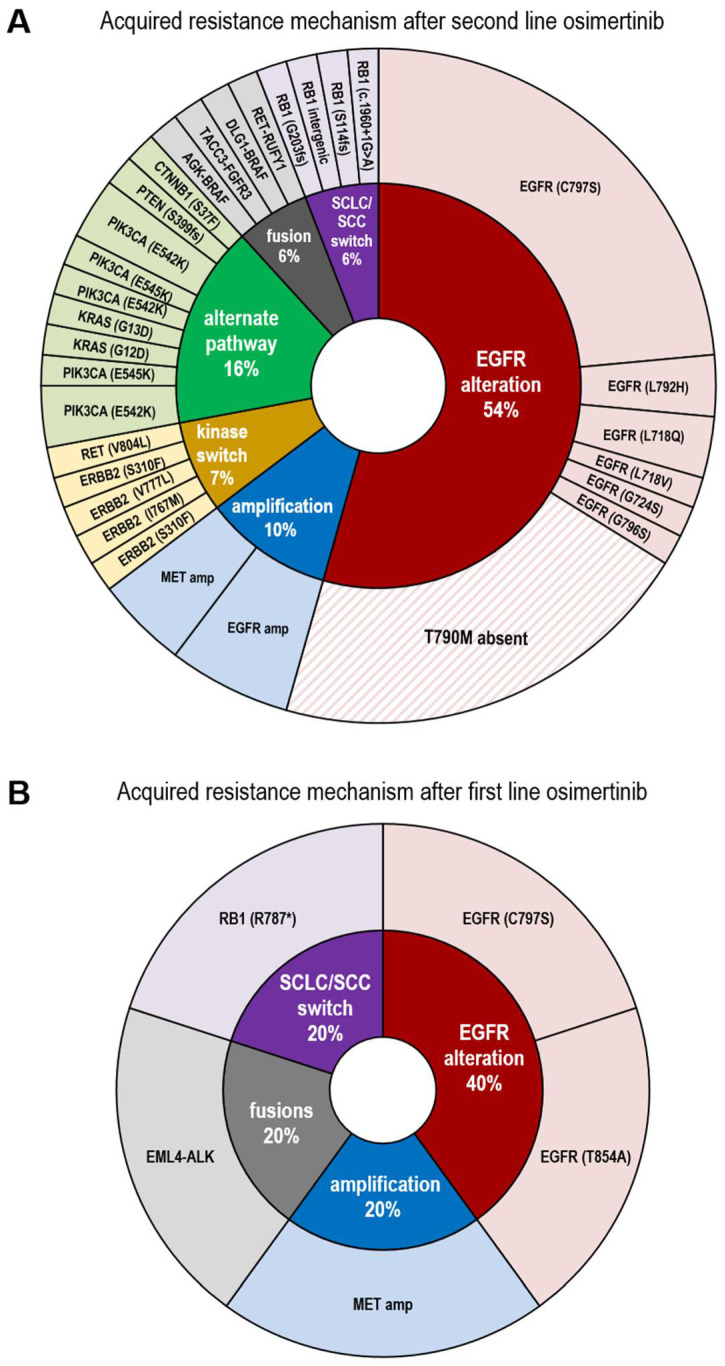
Spectrum of resistance alterations after osimertinib treatment. **(A)** Patients treated sequentially with first or/and second generation TKI followed by osimertinib. CtDNA analysis was performed at time of progression to second line osimertinib. Overall 68 resistance mutations were allocated to 41 patients’ samples. Acquired *EGFR* alterations (*N* = 37, 54%) are colored in red, acquired amplifications (*N* = 7, 10%) in blue, alterations associated with kinase switch in yellow (*N* = 5, 7%), resistance mutations indicative for alternate pathway in green (*N* = 11, 16%), acquired gene fusions in grey (*N* = 4, 6%) and transformation into squamous-cell carcinoma / small cell lung cancer (SCC/SCLC) in purple (*N* = 4, 6%). (**B**) Resistance alterations after first line osimertinib of seven NSCLC patients. Two (40%) *EGFR* resistance alterations (red), one (20%) *MET* amplification (blue), one *EML4-ALK* fusion (grey), One *RB1* mutation indicative for SCC/SCLC switch (purple) respectively. Patients were excluded from analysis in case the initial *EGFR* driver mutation was not detectable in plasma cfDNA.

**Figure 2 cancers-13-02861-f002:**
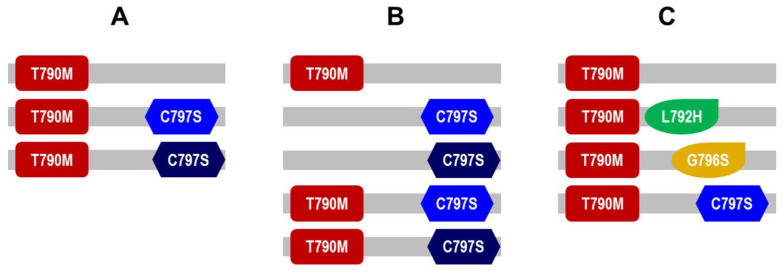
Schematic representation of *EGFR* and *PIK3CA* allelic variants in plasma samples of select patients after osimertinib second line treatment. (**A**) Three distinct alleles: T790M, T790M + C797S (c.2389T > A,) T790M + (C797S c.2390 G > C) (**B**) Five distinct alleles: T790M, C797S (c.2389T > A), C797S (c.2390 G > C), T790M + C797S (c.2389T > A), T790M + C797S (c.2390 G > C) (**C**) Four distinct alleles: T790M, T790M + *PIK3CA* (L792H), T790M + *PIK3CA* (G796S), T790M + C797S (c.2389T > A).

**Table 1 cancers-13-02861-t001:** Overview of patient characteristics.

Variable	Characteristics	Categories	Number (*N*) and Value (%)	
***N*** ** = 56**	Age at diagnosis (years)	Median	63	
		Mean (±SD)	60 (±12)	
		Interquartile range	49.5–71.5	
		<65 years	30	(54%)
		≥65 years	26	(46%)
	Sex	Female	36	(64%)
		Male	20	(36%)
	Smoking status	Current smoker	2	(4%)
		Never smoker	31	(55%)
		Ex-smoker	17	(30%)
		n.d.	6	(11%)
	Histology	Adenocarcinoma	54	(96%)
		Adeno-squamous	1	(2%)
		Large-cell carcinoma	1	(2%)
	UICC stage	IV	55	(98%)
		n.d.	1	(2%)
	Primary *EGFR* mutation type	Exon 19 deletions	40	(71%)
		L858R	13	(23%)
		L861Q	1	(2%)
		E709G, G719S	1	(2%)
		E709V, G719C	1	(2%)
	TKI sequence	sequential therapy	47	(84%)
		osimertinib first line	9	(16%)
***N*** ** = 41**	*EGFR* dependent resistance to first and second generation TKI	T790M	36	(88%)
***N*** ** = 56**	Primary *EGFR* mutation confirmed by liquid biopsy	yes	48	(86%)
		no (after sequential therapy)	6	(11%)
		no (previously treated with osimertinib first line)	2	(4%)
***N*** ** = 41**	Patients resistance under sequential therapy	resistance alteration acquired	33	(80%)
		no resistance alteration	8	(20%)
		absolute number of detected resistance mechanisms	68	
***N*** ** = 7**	Patients resistance under first line osimertinib	resistance alteration acquired	3	(43%)
		no resistance alteration	4	(57%)
		absolute number of detected resistance mechanisms	5	

## Data Availability

The data presented in this study are available on request from the corresponding author upon reasonable request.

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
