# Peer review of "Comparison of Resistance Spectra after First and Second Line Osimertinib Treatment Detected by Liquid Biopsy"

_cancers, 2021, doi:10.3390/cancers13122861_

Round 1
Reviewer 1 Report
The authors reported a comparison of resistance mechanisms appearing during first or second line osimertinib treatment in 56 NSCLC patients using a hybrid capture NGS based liquid biopsy approach. The paper could be considered for publication with some minor revisions.
MINOR REVISIONS
Even if it was described in the text the authors should highlight that most of the reported molecular analyses concern patients after second line osimertinib treatment (47 vs 9 patients) and that in case of alterations suggestive of a SCLC/SCC switch no patient received a confirmatory tissue biopsy.
The authors could add data regarding trials planned to combine EGFR TKIs with anti VEGF treatments ( e.g. RELAY study) or biomarker-directed studies (e.g. ORCHARD trial).
Author Response
Dear Reviewer 1,
we appreciate your comments and do fully agree.
- We have introduced changes to the text to highlight the fact that most patients in our cohort were treated with second line osimertinib. We have modified the sections “Abstract”, “Introduction” and “Material and Methods” accordingly.
- To clarify the limitation that we could not confirm SCLC/SCC transformation histologically, we added this information to all appropriate text passages, specifically in section 4. Resistance spectrum after first line osimertinib treatment as well as in the discussion section.
- You mentioned two highly relevant study concepts within the context of EGFR TKI resistance management, as exemplified by the RELAY and the ORCHARD studies. Both studies, including references were integrated into the discussion section.
Thank you and kind regards.
Reviewer 2 Report
This manuscript describes the results of the HC NGS analysis in a series of patients with non-small cell lung cancer who developed resistance to osimertinib used as second or first line treatment. On the one hand, it highlights the usefulness of HC NGS in liquid biopsy samples stands out as a convenient and non-invasive way to monitor serially the appearance of resistance to treatment. In addition, it is intended to provide a catalogue of molecular defects that identifies the possible causes of resistance to osimertinib and establishes that the spectrum of mechanisms of resistance to osimertinib is greater and heterogeneous compared to the resistance against inhibitors of thymidine kinase of first and second generation. This is of interest since osimertinib will probably see an increase in its use and resistance to its action will become a clinical problem to be addressed.
However, some points need to be clarified by the authors so that the manuscript is more accurate and provides readers with the necessary elements to understand the magnitude of their findings and conclusions.
Major points:
1.
At the end of the Methods section it is included the following paragraph regarding Statisitical issues:
Descriptive statistics were used. Differences between groups were tested with parametric or non-parametric methods depending on the distribution type. Statistical significance assessed by Student t and ANOVA (Alpha level 0.05). Pearson correlation was performed using Microsoft Excel.
However, apart from descriptive statistics of frequencies of molecular defects, no results are shown in the results section regarding any difference between groups or correlations. Authors should include these analysis and comment on them.
2.
2.
There is an apparent disagreement between Figure 1(B) and Section 3.4. describing the resistance spectrum to first line osimertinib.
Figure legend indicates that 7 patients developed resistance alteration to first line osimertinib treatment. However, in Section 3.4 it is stated that only in 3 of 7 patients with characterised EGFR driver mutation were found molecular mechanisms for resistance.
Moreover, percentage figures (20%, 40%) of alterations reflected in the Figure 1(B) do not match with the expected numbers if the number of included subjects was 7 or even 3.
Minor points:
1.
Some abbreviations are used without defining them the first time they are used. Although PFS (progression-free survival) and OS (overall survival) are commonly used in the field of oncology, it would be advisable to define them at their first appearance in the text.
2.
Reference format does not fit the requirements of the journal (full list of authors, page range).
Author Response
Dear Reviewer 2,
thank you for your in depths evaluation and your very valid points of concern. We would like to address them one by one:
- During the early phase of data collecting we indeed intended to provide statistical analysis as described within the material and methods section. As data grew more mature, however we regarded the numbers too low to perform meaningful statistics (nine patients first line and 47 patients second line). We subsequently simply forgot to remove the “statistics draft version” from the materials and methods section. This is very unfortunate as it has evoked confusion and we apologize for this inaccuracy. We have removed the respective text passage from the section Materials and Methods (major point 1).
- We have carefully re-evaluated the data and corrected our mistake in figure 1. Data in figure 1 are now in line with numbers described in the text (major point 2).
- As suggested, we have defined the abbreviations for progression-free survival and overall survival at their first appearance in the text (minor point 1).
- We have adjusted the reference format in accordance with the Journal´s specifications (minor point 2).
Thank you and kind regards.
Reviewer 3 Report
The authors showed that the genetic alteration of ctDNA brought by the application of third generation EGFR TKI, Osimertinib, in NSCLC patients. Followings are my specific comments.
Major
- It is difficult to understand the scientific progress by this work from AURA3 test and FLAURA test. Authors should present about their findings clearly.
- Patient cohort, especially patients applied Osimertinib for the first line treatment, is small. At least, histopathological assessment of representative patients is needed because grade of malignancy is associated with mutations of cancer driver genes.
Minor
- Line 40. Show the long form of PFS.
- Line197. “cDNA” is “ctDNA”?
- Author should check for English and grammar.
Author Response
Dear Reviewer 3,
thank you for your detailed revision, we would like to address your comments one by one:
- We have reviewed the literature regarding the AURA and FLAURA trials and have discussed potential explanations for differences in C797X frequency observed in our data. Importantly in AURA3 patients were tested for resistance alterations in case of progression under osimertinib or therapy discontinuation, while patients of our cohort were clearly defined by clinical progression at the time of liquid biopsy taking. This hypothesis is strengthened by the observation that in AURA3, 60% of patients had no known resistance alteration versus merely 20% in our cohort. Further, we have discussed the potential effect of different degrees of analytical sensitivity between the assays used (major point 1).
- We agree that the cohort treated with osimertinib first line is comparatively small and your concern is in line with reviewer 1. We have introduced changes to the text to highlight the fact that most patients in our cohort were treated with second line osimertinib. We have modified the sections “Abstract”, “Introduction” and “Material and Methods” accordingly and shortened the discussion part to prevent over-interpretation of results derived from a rather limited cohort. Patients from our study were molecularly tested as part of a national lung cancer network NOWEL. This network was created to enable reimbursement of HC NGS diagnostics on blood samples from stage IV NSCLC patients for whom solid tumor tissue could either not be obtained at all, or material was insufficient for molecular testing of all relevant driver mutations. As a result, within the group of patients treated with osimertinib fist line, we unfortunately have no tumor tissue available that we could use to assess mutation status (major point 2).
- We have defined the abbreviations for progression-free survival and overall survival at their first appearance in the text. We have corrected the typing error “cDNA” into “ctDNA”. Furthermore, as suggested by you, we have checked and corrected the manuscript overall for linguistic deficits (minor point).
Thank you and kind regards.
Reviewer 4 Report
The manuscript titled “Comparison of Resistance Spectra After First and Second Line Osimertinib Treatment Detected by Liquid Biopsy” provides important data for deciphering the mechanism of resistance to osimertinib through a hybrid capture (HC) next generation sequencing (NGS) based liquid biopsy approach. This study was well done and the paper was well written overall. The following minor points must be addressed before the manuscript can be suitable for publication.
Comments:
- The authors should indicate what each column means in table1 (including units).
- The authors need to explain more details of Figure 1.
- In 2.2 Liquid Biopsy of materials and methods part, the authors need to mention the numbers that the whole blood was collected by numerous patients.
Author Response
Dear Reviewer 4,
thank you very much for your kind remarks and for your constructive comments.
- We have introduced adequate titles including units to the columns of table 1.
- Legend to figure 1 was rewritten and more information was included.
- In section 2.2 we clarified that for each patient two Streck tubes of whole blood were collected at the time of progression under osimertinib.
Thank you and kind regards.
Reviewer 5 Report
Since you only have a sample size of less than 100, all percentages are only accurate to integer accuracy. I don't believe age should be presented beyond integer accuracy. For age, where you provide the median age, interquartile range should also be provided. This issue applies to text as well as to Figures and Tables.
Statistical analyses: - unclear to me where listed tests were used. Each Figure or Table where statistical evaluation of the data was conducted should indicate which test was used and the P value determined.
Results - sections 3.2 and 3.3 - contains elements of methods that should be relocated to that section of the manuscript.
I have some difficulty will data selection for discussion, since in some instances, the occurrence of the genetic change is relatively rare in a rather small overall sample size.
There is redundancy of results in the Discussion section.
Given the rather small sample size and the relatively recent availability of osimertinib, I have some difficulty in understanding the justification for the rather involved Discussion section provided.
Author Response
Dear Reviewer 5,
we thank you for your constructive and very valid comments.
- We have changed all percentage values to integer accuracy within the figures, tables and within the text. For the parameter “age” we have added the interquartile range to table 1 as suggested.
- During the early phase of data collection we indeed intended to provide statistical analysis as described within the material and methods section. As data grew more mature, however we regarded the numbers too low to perform meaningful statistics (9 patients first line and 47 patients second line). We subsequently forgot to remove the “statistics draft version” from the materials and methods section. This is very unfortunate as it has evoked confusion and we apologize for this inaccuracy. We have removed the respective text passage from the section.
- As you suggested, we have moved parts of sections 3.2 and 3.3 from results to materials and methods.
- We have shortened the discussion at several text passages to reduce redundancy and to prevent over-interpretation of results based on limited patient numbers (especially the osimertinib first line subgroup, n=9). Along those lines, we have introduced changes to the text to highlight the fact that most patients in our cohort were treated with second line osimertinib (with this we are also referring to a similar request by another reviewer). We have modified the sections “Abstract”, “Introduction” and “Material and Methods” accordingly.
Thank you and kind regards.
Round 2
Reviewer 2 Report
The authors addressed properly the points I raised in my report. I think the manuscript can be accepted fro publication in the present form.
Reviewer 3 Report
The authors adequately, if not completely, addressed my concerns. The manuscript is suited for publication in Cancers.
Reviewer 5 Report
Authorss have addressed sufficiently the concerns raised in my original review.